# Temporal Knowledge Graph Reasoning Based on N-tuple Modeling

**Zhongni Hou[1,2], Xiaolong Jin[1,2*], Zixuan Li[2*], Long Bai[2], Saiping Guan[2],**
**Yutao Zeng[3], Jiafeng Guo[1,2] and Xueqi Cheng[1,2]**

[1]School of Computer Science and Technology, University of Chinese Academy of Sciences;
[2]CAS Key Laboratory of Network Data Science and Technology,
Institute of Computing Technology, Chinese Academy of Sciences; [3]Tencent Inc.
{houzhongni18z,jinxiaolong,lizixuan,bailong,guansaiping}@ict.ac.cn
{guojiafeng,cxq}@ict.ac.cn;yutao.zeng@outlook.com

## Abstract

Reasoning over Temporal Knowledge Graphs (TKGs) that predicts temporal facts (e.g., events) in the future is crucial for many applications. The temporal facts in existing TKGs only contain their core entities (i.e., the entities playing core roles therein) and formulate them as quadruples, i.e., (subject entity, predicate, object entity, timestamp). This formulation oversimplifies temporal facts and inevitably causes information loss. Therefore, we propose to describe a temporal fact more accurately as an n-tuple, containing not only its predicate and core entities, but also its auxiliary entities, as well as the roles of all entities. By so doing, TKGs are augmented to N-tuple Temporal Knowledge Graphs (N-TKGs). To conduct reasoning over N-TKGs, we further propose N-tuple Evolutional Network (NE-Net). It recurrently learns the evolutional representations of entities and predicates in temporal facts at different timestamps in the history via modeling the relations among those entities and predicates. Based on the learned representations, reasoning tasks at future timestamps can be realized via task-specific decoders. Experiment results on two newly built datasets demonstrate the superiority of N-TKG and the effectiveness of NE-Net.

## 1 Introduction

Knowledge Graphs (KGs), which represent facts in the form of triples (Bordes et al., 2013; Dettmers et al., 2018), i.e., (subject entity, predicate, object entity), have attracted increasing research attention in recent years. As the validity of facts can change over time, Temporal Knowledge Graphs (TKGs) extend triples into quadruples (Han et al., 2020b; Park et al., 2022), i.e., (subject entity, predicate, object entity, timestamp), to represent temporal facts, such as events. Reasoning over TKGs aims to answer queries about future temporal facts, such as (America, Sanction, ?, 2024-1-10), based on the

---
[*]Corresponding authors.

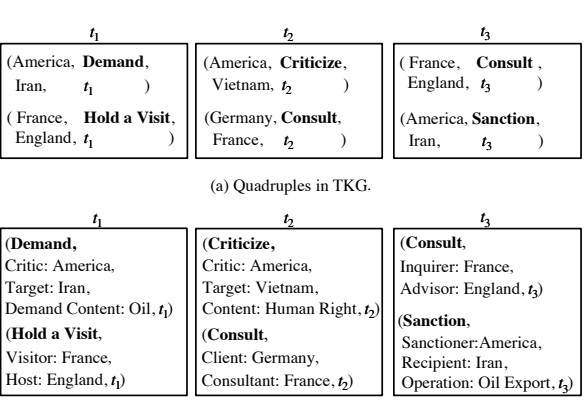

(a) Quadruples in TKG.

(b) N-tuples in N-TKG.

Figure 1: A comparison of quadruples in TKG and n-tuples in N-TKG.

observed history. Such task can be applied to many practical applications, including disaster relief (Signorini et al., 2011), financial analysis (Bollen et al., 2011), etc.

Actually, besides its core entities (i.e., subject entity and object entity), a temporal fact often involves other auxiliary roles and the corresponding arguments (i.e., entities). As illustrated in Figure 1, the "Demand" event involves not only the core roles and entities, but also the auxiliary role "Demand Content" and its corresponding entity "Oil". Moreover, the number of roles in temporal facts may be variable. For example, the "Hold a Visit" event contains two roles, while the "Criticize" event has three. Above all, the existing quadruple-based formulation cannot describe temporal facts accurately, thus limiting the applications of TKGs.

Motivated by these, we propose to describe temporal facts more accurately as n-tuples and correspondingly augment TKGs as N-tuple Temporal Knowledge Graphs (N-TKGs), so as not to cause information loss. Specifically, each n-tuple in N-TKGs is denoted in form of (predicate, $role_1$:$entity_1$, ..., timestamp). For example, the event "Consult" in Figure 1 is described as (Consult, Client: Germany, Consultant: France, Consult Content: Iran, $t_2$).

Similar to that of TKGs, reasoning over N-TKGs is an important task for their practical applications. However, existing methods for the reasoning task on either TKGs with quadruples or static KGs with n-tuples have limitations when facing N-TKGs. In more detail, the reasoning methods for quadruple-based TKGs (Dasgupta et al., 2018; Goel et al., 2020; Jin et al., 2020; Han et al., 2020b; Park et al., 2022) cannot be directly applied to n-tuples and have to take adaptation measures. Those reasoning methods for static n-tuple KGs (Rosso et al., 2020; Guan et al., 2020; Liu et al., 2021; Guan et al., 2019) cannot capture temporal information contained in those facts at different timestamps.

To solve the above problems, we propose a model called N-tuple Evolutional Network (NE-Net) to conduct reasoning tasks over N-TKGs. NE-Net consists of an entity-predicate encoder and task-specific decoders. The entity-predicate encoder is used to learn the evolutional representations of entities and predicates in different temporal facts. Specifically, at each timestamp, it employs an entity-predicate unit to capture the relations among entities and predicates formed upon concurrent facts. It also adopts a core-entity unit to emphasize and more directly model the relations between core entities (in terms of predicates) of individual temporal facts, as such relations contain the most primary information of the facts. Next, an aggregation unit is utilized to integrate the outputs of these two units so as to obtain more accurate representations of entities and predicates. Note that the encoder employs a recurrent mechanism to autoregressively learn the evolutional representations from the facts at temporally adjacent timestamps, and thus implicitly reflects temporal behavioral patterns of entities in their evolutional representations.

NE-Net finally leverages task-specific decoders to conduct different reasoning tasks, namely, predicate reasoning and entity reasoning, respectively, based on the learned representations of entities and predicates. In addition, as there is no N-TKG dataset publicly available, we construct two new datasets, named NWIKI and NICE, to facilitate the research on reasoning over N-TKGs.

In summary, our contributions are as follows: (1) We propose to use n-tuples to describe temporal facts more accurately, and further enhance TKGs as N-TKGs; (2) We propose NE-Net to conduct the reasoning tasks over N-TKGs. It leverages an entity-predicate encoder to learn accurate evolu-

tional representations via capturing the relations among entities and predicates formed upon concurrent facts and simultaneously emphasizing the relations between core entities. NE-Net further utilizes task-specific decoders to address different reasoning tasks; (3) Experiments on the two new datasets demonstrate the superiority of N-TKG and the effectiveness of NE-Net.

## 2 Related Works

**Static N-tuple KG Reasoning.** Static n-tuple KG reasoning aims to infer the missing elements of a given n-tuple. Existing methods can be divided into three categories, namely, hyperplane methods, multi-linear methods and neural methods. Hyperplane methods (Wen et al., 2016; Zhang et al., 2018) project entities into relation hyperplanes to calculate the plausibility scores for n-tuples. Multi-linear methods (Liu et al., 2021) apply the multi-linear product to measure the plausibility scores. Neural methods (Guan et al., 2019; Galkin et al., 2020) leverage CNN or GCN to capture the relatedness scores of role-entity pairs in n-tuples. Particularly, NeuInfer (Guan et al., 2020) and HINGE (Rosso et al., 2020) notice that different elements in n-tuples are of different importance and propose to represent an n-tuple as a main triplet along with auxiliary role-entity pairs. However, the occurrence time of temporal facts is viewed as a common kind of information about the facts and is usually ignored by the above methods (Rosso et al., 2020). Therefore, these methods cannot model the temporal behavioral patterns across adjacent timestamps and are not effective enough to exploit historical data for future predictions.

**TKG Reasoning.** There are two different task settings for TKG reasoning, namely, interpolation and extrapolation (Li et al., 2022c; Han et al., 2021b; Sun et al., 2021; Li et al., 2022b,a). TKG reasoning under the former setting is to infer missing elements of facts at known timestamps (i.e., the history) (Sadeghian et al., 2016; Esteban et al., 2016; Han et al., 2020a; Leblay and Chekol, 2018). Under the interpolation setting, HyTE (Dasgupta et al., 2018) extends the idea of TransH (Wang et al., 2014) and associates each timestamp with a corresponding hyperplane. DE-DistMult (Goel et al., 2020) and DE-SimplE (Goel et al., 2020) both utilize a diachronic embedding for entities and relations, dividing the representations of entities into a static segment and a time-varying seg-

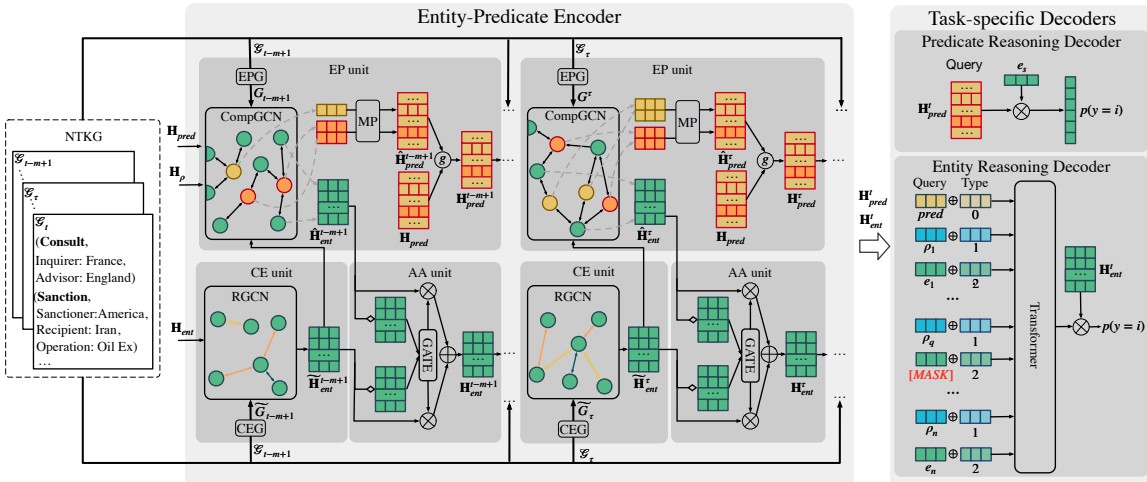

Figure 2: An illustrative diagram of the proposed NE-Net model for N-TKG reasoning.

ment. TKG reasoning under the latter setting, which this paper focuses on, is to infer missing elements of facts at future timestamps (Han et al., 2020b, 2021a; Park et al., 2022). RE-Net (Jin et al., 2020) and REGCN (Li et al., 2021b) both adopt a relation-aware GCN and a recurrent component to capture temporal associations within the history. CyGNet (Zhu et al., 2020) applies a time-aware copy-generation network to model the repetitive pattern of frequently occurring facts. Focusing on the explainable TKG reasoning methods, TITer (Sun et al., 2021) and CluSTeR (Li et al., 2021a) employ reinforcement learning to adaptively search informational facts in the history. More recently, CEN (Li et al., 2022b) utilizes a length-aware CNN to mine complex temporal patterns of different lengths. CENET (Xu et al., 2023) employs the contrastive learning strategy to identify potential entities from historical and non-historical dependency. In the above methods, the facts are all in the form of quadruples. Therefore, they cannot directly model the temporal associations involving auxiliary entities.

## 3    Problem Formulation

**N-TKG.** An N-TKG $\mathcal{G}$ can be formalized as a sequence of KGs with timestamps, i.e., $\mathcal{G} = \{\mathcal{G}_1, \mathcal{G}_2, ..., \mathcal{G}_t, ...\}$. The KG, $\mathcal{G}_t$, at timestamp $t$ can be denoted as $\mathcal{G}_t = (V_{pred}, V_{ent}, V_\rho, F_t)$, where $V_{pred}, V_{ent}, V_\rho, F_t$ are the sets of predicates, entities, roles and temporal facts (hereinafter referred to as facts if not causing any confusion) occurring at timestamp $t$, respectively. Each fact $f \in F_t$ is denoted as $(pred, \rho_1 : e_1, ..., \rho_i : e_i, ..., \rho_n : e_n, t)$, where $pred \in V_{pred}$ is its predicate; each $e_i \in V_{ent}$ ($1 \le i \le n$) is an entity involved in this fact, where

it plays the role as $\rho_i \in V_\rho$; $n$ and $t$ are the total number of role-entity pairs and the timestamp of the n-tuple $f$, respectively. Note that in any fact $f \in F_t$, $e_1$ and $e_2$ are the two core entities, corresponding to its subject and object ones, respectively. Moreover, in this paper, the same type of facts is denoted with the same predicate. Therefore, $pred$ also denotes the type of $f$.

**Predicate reasoning.** It aims to predict the type of the fact (i.e., predicate) that will occur on the given entity $e_s$ at the next timestamp $t+1$. In other words, it aims to answer the query like $(?, -:-, ..., -:e_s, ..., -:-, t+1)$. Here, only the target entity $e_s$ is available and it can be any entity in the fact.

**Entity reasoning.** It aims to predict the unknown entity playing the given role $\rho_q$ in a fact of the type specified by predicate $pred$ at the next timestamp $t+1$. Formally, this task aims to answer the query like $(pred, \rho_1 : e_1, \rho_2 : e_2, ..., \rho_q : ?, ..., \rho_n : e_n, t+1)$. Here, all $\rho_i$ and $e_i$ ($1 \le i \le n$ and $i \ne q$) are given.

## 4    The NE-Net Model

As illustrated in Figure 2, NE-Net consists of an entity-predicate encoder and task-specific decoders. The entity-predicate encoder recurrently models the relations among entities and predicates generated by concurrent facts in the history and outputs the evolutional representations of all entities and predicates. Based on the learned representations, task-specific decoders are adopted to conduct different reasoning tasks.

### 4.1    The Entity-Predicate Encoder

There are mainly two kinds of information in historical facts, namely, the explicit relations among entities and predicates in concurrent facts at the

same timestamp, and the implicit temporal behavioral patterns of entities reflected in different timestamps. The entity-predicate encoder incorporates these two kinds of information into the representations of entities and predicates, via recurrently modeling the KG sequences. In this paper, we consider the history at the latest $m$ timestamps, i.e., $\{\mathcal{G}_{t-m+1}, \mathcal{G}_{t-m+2}, ..., \mathcal{G}_t\}$, where $t$ is the current timestamp.

### 4.1.1 Entity-Predicate Modeling in Concurrent Facts

In this paper, by denoting a temporal fact $f$ as $(pred, \rho_1 : e_1, ..., \rho_i : e_i, ..., \rho_n : e_n, t)$, its predicate $pred$ forms natural relations with its all entities $e_i$ ($1 \leq i \leq n$) specified by $\rho_i$, respectively. Such relations between the entities and predicate preserve the relatively complete information of the fact and are referred to as the entity-predicate relations. Among these relations, those between the predicate and core entities are more important than others, as they reflect the primary information of the fact and should thus be given more attention. Besides these relations within a fact, by sharing the same entity, the predicates and entities in different facts, being concurrent in the same timestamp, may form more complicated structures.

To capture the above relations and structures so as to finally learn the representations of entities and predicates, the encoder first utilizes an Entity-Predicate unit ("EP unit" in Figure 2) to model the relatively complete entity-predicate relations in concurrent facts. It then utilizes a Core-Entity unit ("CE unit" in Figure 2) to highlight the relations between core entities. Finally, the above relational information captured by the EP and CE units is integrated via an Attention-based Aggregation unit ("AA unit" in Figure 2) to obtain more accurate evolutional representations of entities and predicates.

**The Entity-Predicate Unit.** In order to employ the EP unit to model the relatively complete entity-predicate relations in all concurrent facts at timestamp $\tau$ ($t - m \leq \tau < m$), we first need to construct an Entity-Predicate Graph ("EPG" in Figure 2) based on them. To do so, for each fact, we denote its predicate and all entities as the nodes of the EPG and the entity-predicate relations as its edges. As aforesaid, in this paper, the facts of the same type use the same predicate. Therefore, in order to avoid confusion between different facts of the same type, we create the same number of predicate instance nodes in the EPG, each of which corresponds to a specific fact. By so doing, the EPG at timestamp $\tau$ can be formulated as $G_\tau = (V_{ent} \cup \hat{V}_{pred}^\tau, E_\tau)$, where $V_{ent}$, $\hat{V}_{pred}^\tau$ and $E_\tau$ are the sets of entity nodes, predicate instance nodes, and edges, respectively. Here, each predicate instance node is only associated with the entities in the corresponding fact. And, there is a mapping function $\varphi(\cdot) : \hat{V}_{pred}^\tau \rightarrow V_{pred}$, which maps the predicate instance node $v$ to its type $\varphi(v) \in V_{pred}$ to indicate which kind of predicate it belongs to. Each edge $e_{ij} = (v_i, \rho, v_j) \in E_\tau$ links the entity node $v_i$ and the predicate instance node $v_j$ via the corresponding role $\rho$. Note that for those entities and predicates not involved in any fact at timestamp $\tau$, self-loop edges are added.

Upon the EPG, the EP unit works with two steps, namely, the message-passing step and the pred-aggregation step. It finally updates and outputs the entity representation matrix $\hat{\mathbf{H}}_{ent}^\tau$ and the predicate representation matrix $\mathbf{H}_{pred}^\tau$ at timestamp $\tau$.

The message-passing step employs a CompGCN (Vashishth et al., 2019) with $\omega_1$ layers, to update the representations of entities and predicate instances. Specifically, the representation of the node $v$ at layer $l \in [0, \omega_1 - 1]$ is obtained as follows:

$$\boldsymbol{h}_v^{\tau, l+1} = \psi \left( \sum_{(u, \rho, v) \in G_\tau} \mathbf{W}_0^l (\boldsymbol{h}_u^{\tau, l} + \boldsymbol{\rho}) + \mathbf{W}_1^l \boldsymbol{h}_v^{\tau, l} \right), \quad (1)$$

where $\boldsymbol{h}_v^{\tau, l}$ denotes the representation of node $v$ obtained after $l$ layers; $\boldsymbol{\rho}$ denotes the representation of the role $\rho$; $\mathbf{W}_0^l$ and $\mathbf{W}_1^l$ are the parameters in the $l$-th layer; $\psi(\cdot)$ is an activation function. The representations of predicate instance nodes at the first layer are obtained by looking up the input evolutional representation matrix of predicates $\mathbf{H}_{pred}^{\tau-1}$ according to its type. In particularly, at the first timestamp $t - m + 1$, the randomly initialized predicate representation matrix $\mathbf{H}_{pred}$ is used as the input. The representations of entity nodes at the first layer are calculated by the CE unit, which will be introduced in the following. The representations of roles are obtained from the role representation matrix $\mathbf{H}_\rho$, which is randomly initialized and shared across timestamps.

At timestamp $\tau$, the pred-aggregation step updates the evolutional representation of a predicate $\boldsymbol{h}_{pred}^\tau$ by aggregating the information of all related nodes. For a predicate $pred$, after the message passing step, its different predicate instance nodes have different representations due to the different

involving entities. NE-Net utilizes the mean pooling (MP) operation to summarize the information of its predicate instance nodes $\hat{\boldsymbol{h}}_{pred}^{\tau}$:

$$\hat{\boldsymbol{h}}_{pred}^{\tau} = MP \left( \sum_{v \in \{v | \varphi(v) = pred\}} \boldsymbol{h}_{v,pred}^{\tau} \right). \quad (2)$$

Besides, for a predicate $pred$, to preserve the inherent semantic information, NE-Net further incorporates the initial representation $\boldsymbol{h}_{pred}$, into its evolutional representation $\boldsymbol{h}_{pred}^{\tau}$ at timestamp $\tau$ as:

$$\boldsymbol{h}_{pred}^{\tau} = g(\hat{\boldsymbol{h}}_{pred}, \boldsymbol{h}_{pred}) = \mathbf{W}_2(\hat{\boldsymbol{h}}_{pred}^{\tau} || \boldsymbol{h}_{pred}), \quad (3)$$

where $\mathbf{W}_2$ is a learnable parameter matrix and $||$ denotes the concatenation operation.

**The Core-Entity Unit.** As aforementioned, the most important information in a fact is the relation between its two core entities in terms of the predicate, which is more important than other relations. To emphasize these relations between core entities more directly, this unit views predicates as edges, and constructs the Core Entity Graph ("CEG" in Figure 2) for concurrent facts at timestamp $\tau$. Specifically, the CEG at timestamp $\tau$ can be formulated as $\widetilde{G}_\tau = (V_{ent}, \widetilde{E}_\tau)$, where $V_{ent}$ and $\widetilde{E}_\tau$ are the sets of core entity nodes and edges, respectively. Each edge $e_{ik} = (v_i, pred, v_k) \in \widetilde{E}_\tau$ links the two core entity nodes $v_i$ and $v_k$ corresponding to a certain fact occurring at timestamp $\tau$ via its predicate $pred$ of an n-tuple. Similarly, for those entities not involved in any fact, self-loop edges are added.

Based on the constructed CEG $\widetilde{G}_\tau$ and the input representation matrix of entities $\mathbf{H}_{ent}^{\tau-1}$ at timestamp $\tau - 1$, this unit leverages an RGCN (Schlichtkrull et al., 2018) with $\omega_2$ layers to encode the relations between core entities into the updated entity representation matrix $\widetilde{\mathbf{H}}_{ent}^{\tau}$:

$$\widetilde{\mathbf{H}}_{ent}^{\tau,l+1} = RGCN(\widetilde{\mathbf{H}}_{ent}^{\tau,l}, \widetilde{G}_\tau), \quad (4)$$

where $\widetilde{\mathbf{H}}_{ent}^{\tau,l}$ denotes the updated entity representation matrix obtained at the $l$-th layer. The representations of entity nodes at the first layer are obtained by looking up the input evolutional representation matrix of entities $\mathbf{H}_{ent}^{\tau-1}$. For the first timestamp $t - m + 1$, the randomly initialized entity representation matrix $\mathbf{H}_{ent}$ is used as the input. As the relations between core entities maintain the most primary information of temporal facts, NE-Net directly uses the output of this unit, i.e., $\widetilde{\mathbf{H}}_{ent}^{\tau}$, as the input entity representation matrix of the entity-predicate unit to emphasizing such information.

**The Attention-based Aggregation Unit.** To integrate the information in the above two kinds of relations, i.e., $\widetilde{\mathbf{H}}_{ent}$ and $\hat{\mathbf{H}}_{ent}$, the attention-based aggregation unit is utilized to learn importance weights of the two kinds of information, and adaptively combine them in order to obtain the final evolutional representations of entities $\mathbf{H}_{ent}^{\tau}$ at timestamp $\tau$. Specifically, take the entity $i$ as an example, its evolutional representation $\boldsymbol{h}_{i,ent}^{\tau}$ is calculated as follows:

$$\boldsymbol{h}_{i,ent}^{\tau} = \tilde{a}_{i,ent}^{\tau}\tilde{\boldsymbol{h}}_{i,ent}^{\tau} + \hat{a}_{i,ent}^{\tau}\hat{\boldsymbol{h}}_{i,ent}^{\tau}, \quad (5)$$

where $\tilde{\boldsymbol{h}}_{i,ent}$ and $\hat{\boldsymbol{h}}_{i,ent}^{\tau}$ are the updated representations of entity $i$ outputted by the above two units, $\tilde{a}_{i,ent}$ and $\hat{a}_{i,ent}$ measure the importance weights of two representations. Taking the representation $\tilde{\boldsymbol{h}}_{i,ent}^{\tau}$ as an example, NE-Net first gets its importance value $\tilde{w}_{i,ent}^{\tau}$:

$$\tilde{w}_{i,ent}^{\tau} = f(\tilde{\boldsymbol{h}}_{i,ent}^{\tau}) = \boldsymbol{q} tanh(\mathbf{W}_3 \tilde{\boldsymbol{h}}_{i,ent}^{\tau} + \boldsymbol{b}), \quad (6)$$

where $\mathbf{W}_3$ is the weight matrix, $\boldsymbol{q}$ is the learnable importance gate. Similarly, $\hat{w}_{i,ent}^{\tau} = f(\hat{\boldsymbol{h}}_{i,ent}^{\tau})$.

After obtaining the importance values of each representation, NE-Net normalizes them to get the final importance weights, i.e., $\tilde{a}_{i,ent}^{\tau}$ and $\hat{a}_{i,ent}^{\tau}$, via the softmax function.

### 4.1.2 Entity-Predicate Modeling across Different Timestamps

The temporal patterns hidden in the historical facts of a specific entity implicitly reflect its behavioral trends and preferences. As the temporal facts at a specific timestamp are already modeled in the evolutional representations of entities and predicates via the above three units, NE-Net directly uses the evolutional representations of entities and predicates learned from timestamp $\tau - 1$, namely, $\mathbf{H}_{ent}^{\tau-1}$ and $\mathbf{H}_{pred}^{\tau-1}$, as the input of interaction modeling step for concurrent facts. By recurrently modeling on KGs at adjacent timestamps, the temporal behavioral patterns of entities can be modeled.

### 4.2 Task-specific Decoders

Based on the evolutional representations of entities and predicates at timestamp $t$, NE-Net utilizes task-specific decoders to conduct different types of reasoning tasks.

### 4.2.1 Predicate Reasoning Decoder

Give the query of predicate reasoning $(?, -:-, ..., -:$ $e_s, ..., -:-, t+1)$, NE-Net multiplies the evolutional representation of $e_s$ at timestamp $t$ with the evolutional representations of predicates $\mathbf{H}_{pred}^t$,

to generate the conditional probability vector $\boldsymbol{p}(pred|e_s, \mathcal{G}_{1:t})$:

$$\boldsymbol{p}(pred|e_s, \mathcal{G}_{1:t}) = \sigma(\mathbf{H}_{pred}^\tau \boldsymbol{e}_s^t), \quad (7)$$

where $\sigma(\cdot)$ is the sigmoid function.

### 4.2.2 Entity Reasoning Decoder

To deal with queries having a varied number of role-entity pairs and capture the relations among elements within the query, NE-Net reorganizes role-entity pairs in a query as a sequence and then designs a Transformer-based decoder for the entity reasoning task. Specifically, given a query $(pred, \rho_1 : e_1, ..., \rho_q :?, ..., \rho_n : e_n, t+1)$, NE-Net first replaces the missing entity with a special token [MASK], and linearize the query as: $X = (pred, \rho_1, e_1, ..., \rho_q, [MASK], ..., \rho_n, e_n)$. To identify which kind of element the model is dealing with, each element $x_i \in X$ is assigned with a type $type_i \in \{0, 1, 2\}$, where 0 represents a predicate, 1 represents a role, and 2 represents an entity. The input representation of the $i$-th element in the sequence, i.e., $\boldsymbol{h}_i^0$, is calculated by:

$$\boldsymbol{h}_i^0 = \boldsymbol{x}_i^t + \boldsymbol{type}_i, \quad (8)$$

where $\boldsymbol{x}_i^t$ is generated by looking up the evolutional representation matrices of entities and predicates, i.e., $\mathbf{H}_{ent}^t$ and $\mathbf{H}_{pred}^t$, and the role representation matrix, i.e., $\mathbf{H}_\rho$; $\boldsymbol{type}_i$ is obtained by looking up the learnable type representation matrix $\mathbf{H}_{type}$. Then, all input representations are fed into a stack of $L$ successive Transformer blocks (Vaswani et al., 2017). Based on the output of Transformer, the final representation of [MASK], i.e., $\boldsymbol{h}_q^L$, can be obtained, and is multiplied with the evolutional representations of entities $\mathbf{H}_{ent}^t$ to obtain a probability vector over all entities.

$$\boldsymbol{p}(e|pred, \rho_1, e_1, ..., \rho_q, ..., \rho_n, e_n, \mathcal{G}_{1:t}) = \sigma(\mathbf{H}_{ent}^t \boldsymbol{h}_q^L), \quad (9)$$

where $\sigma(\cdot)$ is the sigmoid function.

### 4.3 Model Learning

Given a fact $f = (pred, \rho_1 : e_1, ..., \rho_n : e_n, t+1) \in \mathcal{F}_{t+1}$, let $e$ denotes any entity in $f$, $T$ denotes the number of timestamps in the training set. The objective of the predicate reasoning task is to minimize the following cross-entropy loss:

$$L_{pred} = -\sum_{t=0}^{T-1} \sum_{f \in \mathcal{F}_{t+1}} \sum_{i=0}^{|V_{pred}|-1} y_{t+1,i}^{pred} \log p_i(pred|e, \mathcal{G}_{1:t}), \quad (10)$$

where $y_{t+1,i}^{pred}$ is a 0/1 value, denoting whether the $i$-th predicate occurs on $e$ at timestamp $t+1$,

$p_i(pred|e, \mathcal{G}_{1:t})$ represents the probability score of the $i$-th predicate.

Similarly, the objective of the entity reasoning task is to minimize the following loss:

$$L_{ent} = -\sum_{t=0}^{T-1} \sum_{f \in \mathcal{F}_{t+1}} \sum_{i=0}^{|V_{ent}|-1} y_{t+1,i}^{ent} \log p_i(e| \\ pred, \rho_1, e_1, ..., \rho_q, ..., \rho_n, e_n, \mathcal{G}_{1:t}), \quad (11)$$

## 5 Experiments

### 5.1 Experimental Setup

#### 5.1.1 Datasets

Since there is no N-TKG dataset available for experiments, we build two datasets, namely NWIKI and NICE.

About NWIKI: (1) We downloaded the Wikidata dump dump[1] and extracted the facts which contain the timestamp information and involve human entities; (2) To extract n-tuple facts, we selected predicates with more than 3 role types and retained their corresponding facts; (3) We filtered out entities of low frequency, as their behaviors are difficult to learn and predict for most data-driven based methods. Correspondingly, facts involving these entities were also filtered out; (4) The fact related to the "Position Held" predicate comprise over 50% of the dataset. However, the filtering operation in the previous step could impair connectivity. To maintain a robust connection with other predicates, we retained facts that are associated with other predicates and share core entities with the facts related to the "Position Held" predicate; (5) Following RE-NET (Jin et al., 2020), all facts were split into the training set, the validation set, and the test set by a proportion of 80%:10%:10% according to the time ascending order.

About NICE: (1) We collected the raw event records from ICEWS [2] from Jan 1, 2005 to Dec 31, 2014; (2) From these raw event records, we extracted core entities, timestamps, and partial auxiliary information, i.e., the occurrence places. (3) We observed that the predicate names contain additional auxiliary information. For example, the predicate names "Cooperate Economically" and "Engage in Judicial Cooperation" contain the specified cooperation aspects of the general predicate "Engaged in Cooperation", namely "Economic" and

---

[1] https://archive.org/details/wikibase-wikidatawiki-20171120

[2] https://dataverse.harvard.edu/dataverse/icews

| Dataset | $|V_{pred}|$ | $|V_{ent}|$ | #Train | | | #Valid | | | #Test | | | Timestamps | Time Interval |
|---------|------|------|--------|------|---------|--------|--------|---------|--------|--------|---------|------------|---------------|
| | | | Binary | Nary | Overall | Binary | Nary | Overall | Binary | Nary | Overall | | |
| NWIKI | 22 | 17,481 | 20,686 | 87,711 | 108,397 | 1,847 | 12,523 | 14,370 | 2,438 | 13,153 | 15,591 | 205 | 1 year |
| NICE | 20 | 10,860 | 46,176 | 322,692 | 368,868 | 5,395 | 40,907 | 5,268 | 40,891 | 46,302 | 46,159 | 4,017 | 24 hours |

Table 1: The statistics of the two proposed datasets, NWIKI and NICE.

| Model | NWIKI | | | NICE | | |
|-------|-----------|--------|-------|-----------|--------|-------|
| | Precision | Recall | F1 | Precision | Recall | F1 |
| MLkNN | 53.20 | 60.87 | 55.26 | 6.16 | 25.33 | 8.57 |
| BRkNN | 54.07 | 61.62 | 56.11 | 5.91 | 27.23 | 8.45 |
| MLARAM | 53.50 | 61.14 | 55.55 | 8.48 | 45.52 | 12.23 |
| DNN | 47.72 | 54.77 | 49.65 | 10.56 | 60.36 | 15.45 |
| T-GCN | 73.20 | 75.27 | 72.04 | 16.70 | 39.47 | 27.48 |
| RENET | 76.17 | 93.60 | 83.30 | **43.62** | 40.20 | 38.82 |
| NE-Net | **78.34** | **94.86** | **84.29** | 32.43 | **85.04** | **45.78** |

Table 2: Experimental results on predicate reasoning.

"Militarily", respectively. To obtain such auxiliary information, we designed rule templates to extract them from the specified predicate names and merged these specified predicates into the general ones. (4) Considering the absence of role information in the raw event records, we manually assigned role names to each predicate. (5) Similar to NWIKI, NICE was also split into the training set, the validation set, and the test set by a proportion of 80%:10%:10%.

### 5.1.2 Evaluation Metrics

We employ F1, recall, and precision as metrics for the predicate reasoning task. We adopt Hits@{1, 3, 10} and MRR as metrics for the entity reasoning task. Following Han et al. (2021a), we report the results under the time filter setting.

### 5.1.3 Baselines

For the predicate reasoning task, we compare NE-Net with temporal reasoning models, including RENET (Jin et al., 2019) and T-GCN (Zhao et al., 2019). Following Deng et al. (2019), we further compare NE-Net with DNN, MLkNN (Zhang and Zhou, 2007), BRkNN (Spyromitros et al., 2008), MLARAM (Deng et al., 2019). These four models are simple DNN or KNN-based methods, which use basic count features to conduct reasoning tasks.

For the entity reasoning task, we compare NE-Net with static n-tuple KG reasoning methods and TKG reasoning methods. The static n-tuple KG reasoning methods include: NALP (Guan et al., 2019), NeuInfer (Guan et al., 2020), HypE (Fatemi et al., 2021), HINGE (Rosso et al., 2020), Hy-Transformer (Yu and Yang, 2021), RAM (Liu et al., 2021). Since these methods are not able to model the temporal information, we construct a cumulative graph for all the training facts. The TKG rea-

soning methods include DE-DistMult (Goel et al., 2020), DE-SimplE (Goel et al., 2020), HyTE (Dasgupta et al., 2018), RE-NET (Jin et al., 2020), CyGNet (Zhu et al., 2020), REGCN (Li et al., 2021b), GHT (Sun et al., 2022), CEN (Li et al., 2022b) and TiRGN (Li et al., 2022a). To facilitate this kind of methods to N-TKG, we convert the historical n-tuples into a KG sequence. In each KG, we view both entities and predicate as nodes, and link entity nodes with fact nodes via roles. Such KG sequence is taken as the input of TKG reasoning methods.

### 5.1.4 Implementation Details

For the evolution encoder, the history length $m$ on two datasets is set to 2; the number of Transformer blocks $L$ for NICE and NWIKI is set to 1 and 2, respectively. For all datasets, the numbers, $\omega_1$ and $\omega_2$, of GCN layers in the entity-predicate and core-entity units are set to 4 and 2, respectively; the number of self-attention heads is set to 4.

## 5.2 Experimental Results

### 5.2.1 Results on Predicate Reasoning

To analyze the effectiveness of NE-Net on predicate reasoning task, we compare NE-Net with baselines on both datasets. The results are presented in Table 2. It is shown that NE-Net outperforms baselines across all datasets in terms of F1 and Recall. Notably, the counting-based methods have the worst performance on both datasets, indicating the importance of modeling the relations among entities and predicates at different timestamps in the predicate reasoning task. It can be particularly observed that NE-Net shows lower performance on the precision metric in NWIKI, as compared with RENET. This is because RENET focuses on a target entity to retrieve historical facts, enabling it to utilize a longer history, while NE-Net solely considers the history in the latest timestamps.

### 5.2.2 Results on Entity Reasoning

To investigate the effectiveness of NE-Net, we divide the test set into core and auxiliary categories (denoted as C and AUX) based on the role to be predicted, and conduct the entity reasoning task

Table 3:

| Model | NWIKI | | | | | | | | NICE | | | | | | | |
|---|---|---|---|---|---|---|---|---|---|---|---|---|---|---|---|---|
| | H@1 | | H@3 | | H@10 | | MRR | | H@1 | | H@3 | | H@10 | | MRR | |
| | C | AUX | C | AUX | C | AUX | C | AUX | C | AUX | C | AUX | C | AUX | C | AUX |
| NALP | 10.59 | 7.54 | 10.59 | 10.84 | 22.52 | 15.17 | 14.86 | 10.25 | 14.66 | 39.82 | 26.17 | 49.48 | 43.40 | 59.87 | 23.96 | 46.92 |
| NeuInfer | 19.83 | 6.07 | 25.22 | 8.62 | 28.94 | 11.91 | 23.08 | 8.14 | 13.77 | 14.48 | 28.35 | 18.30 | 47.56 | 29.59 | 24.78 | 19.32 |
| HINGE | 19.10 | 10.85 | 23.47 | 14.05 | 25.91 | 19.88 | 21.74 | 13.83 | 2.92 | 15.88 | 21.04 | 27.85 | 42.83 | 43.52 | 16.01 | 24.89 |
| RAM | 31.42 | 22.68 | 33.36 | 26.01 | 34.36 | 28.14 | 32.63 | 24.72 | 8.37 | 14.48 | 16.41 | 23.26 | 27.13 | 47.64 | 14.38 | 23.45 |
| HypE | 24.91 | 19.83 | 25.39 | 19.95 | 25.75 | 19.98 | 25.26 | 19.91 | 19.16 | 49.88 | 37.22 | 74.31 | 56.33 | 84.89 | 31.50 | 63.35 |
| Hy-Transformer | 33.40 | 18.36 | 35.85 | 23.60 | 37.84 | 27.60 | 34.97 | 21.66 | 28.51 | 61.02 | 44.49 | 82.46 | 61.11 | 91.38 | 39.47 | 72.71 |
| DE-DistMult | 11.44 | - | 16.10 | - | 18.85 | - | 14.17 | - | 8.61 | - | 18.41 | - | 33.59 | - | 16.75 | - |
| DE-SimplE | 10.87 | - | 16.41 | - | 19.14 | - | 13.87 | - | 11.53 | - | 21.86 | - | 34.80 | - | 19.30 | - |
| HyTE | 17.55 | - | 29.99 | - | 35.33 | - | 23.88 | - | 2.35 | - | 21.82 | - | 39.02 | - | 15.15 | - |
| RENET | 33.56 | - | 38.41 | - | 41.28 | - | 36.57 | - | 33.43 | - | 47.77 | - | 63.06 | - | 43.32 | - |
| CyNet | 44.12 | - | 64.71 | - | 67.65 | - | 53.12 | - | 26.61 | - | 41.63 | - | 56.22 | - | 36.81 | - |
| REGCN | 46.25 | - | 65.13 | - | 72.31 | - | 56.78 | - | 37.33 | - | 53.85 | - | 68.27 | - | 48.03 | - |
| GHT | 30.71 | - | 37.78 | - | 39.94 | - | 34.57 | - | 26.61 | - | 41.63 | - | 56.22 | - | 36.81 | - |
| CEN | 30.28 | - | 45.20 | - | 61.04 | - | 40.61 | - | 33.32 | - | 49.29 | - | 64.65 | - | 43.98 | - |
| TiGRN | 50.61 | - | 68.24 | - | 81.13 | - | 61.10 | - | 34.82 | - | 51.54 | - | 66.47 | - | 45.66 | - |
| NE-Net | **66.87** | **46.45** | **76.08** | **66.01** | **80.29** | **77.32** | **72.03** | **57.68** | **38.36** | **68.55** | **54.18** | **88.16** | **69.99** | **94.61** | **48.98** | **79.06** |

Table 3: Experimental results on the entity reasoning task for predicting core and auxiliary entities, respectively.

on two datasets. Comprehensive results on these two categories are presented in Table 3. Especially, we only report the results of TKG reasoning methods on the core category, as they only focus on quadruples.

As shown in Tables 3, NE-Net consistently significantly outperforms static n-tuple KG reasoning methods across all metrics on both datasets. This can be attributed to the capacity of NE-Net to model the rich relations among entities and predicates at different timestamps. On the core category, we observe that TKG reasoning methods under the interpolation setting, perform worse than NE-Net. These methods focus on predicting facts occurring at known timestamps, and can not capture the relations between entities and predicates within newly-emerged temporal facts. Moreover, NE-Net shows better performance than TKG reasoning methods under the extrapolation setting. Different from these methods, NE-Net learns the representations of the predicates as they dynamic evolve, and emphasizes the relations between core entities within facts containing various numbers of auxiliary entities. Besides, NE-Net can capture information of elements within queries. In short, these results convincingly suggest that NE-Net is effective in conducting the reasoning task over N-TKGs.

## 5.3 Ablation Study

To study the effectiveness of each module of NE-Net, we conduct ablation studies on all datasets. The results are summarized in Table 4.

It can be observed that removing the core-entity unit (denoted as -CE) results in worse performance on both datasets, which illustrates that additionally capturing the relations between core entities can help NE-Net learn more precise representa-

| Dataset | | -CE | -AA | -PredAgg | -Trans | NE-Net |
|---|---|---|---|---|---|---|
| NWIKI | C | 66.25 | 59.47 | 70.94 | 57.02 | **72.03** |
| | AUX | 53.97 | 40.69 | 56.52 | 48.03 | **57.68** |
| | All | 59.23 | 48.73 | 62.69 | 51.88 | **63.82** |
| NICE | C | 47.21 | 47.86 | 48.71 | 43.74 | **48.98** |
| | AUX | 78.34 | 78.13 | 78.78 | 75.58 | **79.06** |
| | All | 64.16 | 64.34 | 65.09 | 61.07 | **65.37** |

Table 4: MRR results by different variants of NE-Net.

| Dataset | | 0% | 30% | 60% | 100% |
|---|---|---|---|---|---|
| NWIKI | C | 48.24 | 67.85 | 71.25 | **72.03** |
| | AUX | 10.44 | 34.39 | 52.41 | **57.68** |
| | All | 26.62 | 48.71 | 60.47 | **63.82** |
| NICE | C | 47.65 | 47.91 | 48.37 | **48.98** |
| | AUX | 77.86 | 78.09 | 78.47 | **79.06** |
| | All | 64.10 | 64.34 | 64.76 | **65.36** |

Table 5: MRR results of NE-Net modeling different ratios of auxiliary information.

tions. Notably, -CE has a more significant impact on the core category when compared with that on the auxiliary category, as CEGs predominantly emphasize entities playing core roles in n-tuples. Furthermore, -CE has a greater performance drop on NWIKI in comparison to NICE. Notably, NWIKI has a higher proportion of facts that involve three or more role-entity pairs (46.32%), as compared to NICE (30.97%). As a result, NWIKI exhibits richer relations among entities and predicates. -CE makes it more difficult for the model to distinguish the most important information contained in the relations among core entities in NWIKI.

To verify the necessity of the attention-based aggregation unit (denoted as -AA), we simply use the max pooling operation to integrate the outputs of the entity-predicate unit and the core-entity unit. It can be seen that -AA yields worse results compared to NE-Net, which demonstrates the necessity of adaptively integrating the information learned from the entity-predicte unit and the core-entity unit. Also, the removal of the pred-aggregation

| History at $t-1$ | History at $t$ | Query at $t+1$ | Answer |
|---|---|---|---|
| **(Disapprove,** 
 Opponent: Barack Obama, 
 Proponent: Head of Government (Kenya), 
 Disapprove Way: Denounce, 
 Place: Kenya) | **(Consult,** 
 Consulter: Barack Obama, 
 Consulted: Mulatu Teshome, 
 Consult Way: Engage in Negotiation, 
 Place: Ethiopia) | **(Engage in Diplomatic Cooperation,** 
 Cooperator: Barack Obama, 
 Cooperator: ? 
 Cooperation way: Praise or Endorse) | *Ethiopia* |

| History at $t-1$ | History at $t$ | Query at $t+1$ | Answer |
|---|---|---|---|
| **(Express Intent to Cooperate,** 
 Volunteer: Iran, 
 Cooperation Target: South Africa, 
 Cooperate Content: Engage in Diplomatic Cooperation) 
 **(Express Intent to Cooperate,** 
 Volunteer: Iran, 
 Cooperation Target: Yemen, 
 Cooperate Content: Provide Humanitarian Aid) | **(Engage in Diplomatic Cooperation,** 
 Cooperator: Iran, 
 Cooperator: South Africa, 
 Cooperation way: Sign Formal Agreement) | **(Provide Aid,** 
 Provider: Iran, 
 Recipient: ?; 
 Aid Content: Humanitarian Aid) | *Yemen* |

Table 6: Case studies on the predictions of NE-Net.

operation (denoted as -PredAgg) also results in worse performance than NE-Net, which demonstrates that aggregating the influence of predicate instances and the inherent representations can help learn better representations of predicates. To verify the necessity of the Transformer-based decoder on the entity reasoning task, we replace this decoder with ConvTransE (Dettmers et al., 2018), denoted as -Trans. It can be seen that removing the transformer decoder leads to a decrease in performance, which demonstrates the effectiveness of our entity reasoning decoder.

### 5.4 Detailed Analysis

To illustrate the superiority of N-TKGs, we investigate the performance of NE-Net with different ratios of auxiliary information. We provide NE-Net with histories containing varying ratios (0%, 30%, 60%,100%) of auxiliary information. Here, the auxiliary information in the history is randomly selected, and we run NE-Net three times and report the averaged results.

From Table 5, it can be observed that the performance of NE-Net is positively correlated with the amount of auxiliary information utilized. This suggests that providing more precise descriptions of temporal facts can enhance the performance of TKG reasoning, thereby demonstrating the superiority of the proposed N-TKGs. Additionally, we notice that the ratio of auxiliary information has a greater impact on NWIKI than NICE. Actually, the types of auxiliary roles in NICE are limited, while in NWIKI, there are more diverse types of auxiliary roles, including occupation, employer, winners, and more. These richer types of auxiliary roles in NWIKI can better help with predictions.

### 5.5 Case Studies

We present two cases in Table 6 where NE-Net correctly predicts the answer entity. In this table, $t-1$ and $t$ represent historical timestamps, while $t+1$ represents the query timestamp. In the first case, the correct answer to the query plays as an auxiliary entity in the latest historical fact. It is hard for the model to predict the correct answer without such auxiliary information. In the second case, the auxiliary entity "Humanitarian Aid" helps the model establish a connection between the query and the most informative historical fact, and thereby enabling more accurate predictions.

### 6 Conclusions

In this paper, we proposed to utilize n-tuples to represent temporal facts more precisely, and correspondingly enhanced TKGs as N-TKGs. We further introduced a model called NE-Net, to conduct reasoning over N-TKGs. NE-Net learns evolutional representations of entities and predicates, via modeling the relations among entities and predicates, and highlighting the relations among core entities. Further, it adopts task-specific decoders to conduct different reasoning tasks. Experimental results on two new datasets show the superiority of N-TKG and the effectiveness of NE-Net.

### 7 Acknowledgments

The work is supported by the National Natural Science Foundation of China under grant U1911401, the National Key Research and Development Project of China, the JCJQ Project of China, Beijing Academy of Artificial Intelligence under grant BAAI2019ZD0306, and the Lenovo-CAS Joint Lab Youth Scientist Project. We thank anonymous reviewers for their insightful comments and suggestions.

## 8 Limitations

The proposed NE-Net model cannot handle temporal facts that involve roles with multiple entities, such as the temporal fact "(Attack, Attacker: {$entity_1$, $entity_2$}, Victim: {$entity_3$, $entity_4$}, 2026-10-01)". Furthermore, it is unable to model long history, as the use of deep GCN layers can bring the over-smoothing problem.

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
