# OpenReview forum: "Temporal Knowledge Graph Reasoning Based on N-tuple Modeling"
_EMNLP/2023/Conference — EMNLP 2023 Findings_

### Official Review · Reviewer_ivcT · 2023-08-01

**Typos Grammar Style And Presentation Improvements:** In Table 3, CyGNet, “G” missed.
**Soundness:** 2

**Excitement:**

3: Ambivalent: It has merits (e.g., it reports state-of-the-art results, the idea is nice), but there are key weaknesses (e.g., it describes incremental work), and it can significantly benefit from another round of revision. However, I won't object to accepting it if my co-reviewers champion it.

**Missing References:**

N/A

**Paper Topic And Main Contributions:**

This paper addresss the problem of information loss in traditional temporal knowledge graphs due to the simple quadruple form of facts representation and attempt to model dynamic facts with n-tuples associated with timestamps yielding N-TKGs. To infer missing fasts in N-TKGs, this paper also proposes a N-TKGs reasoning model, NE-Net, which incorpoorates three units in the encoding peroid to capture interactions between entities involed in various facts and facts at adjacent timestamps. In the decoding stage, the model uses different task-specific decoders to deal with tasks including relation predictions and entity predictions. This paper also provides two N-TKGs datasets built uppon Wikidata and ICEWS for evaluations. Experimental results on the two newly built datatsets show that the proposed method is superior to several baseline models of TKG and NKG reasoning.

**Questions For The Authors:**

1.As mentioned in weakness, what is a predicate instance and predicate type?

2. As mentioned in weakness, why the time window m is set to only 2 in experiments?

3. In line 500, the authors mentioned they use time filter protocol for evaluation. I’d like to know more about the time-aware filtering protocol implemented in NTKGs. For example, in core entity prediction, does it filter out facts with the same core entity, predicate and timestamps or it filter out facts whose entities, predicate and timestamp are completely the same as facts in datasets? How it is implemented during AUX prediction?

4. Refer to weakness (2), why some recent and competitive models are not evaluated? Especially, the improvements on NICE are marginal.

5. Refer to weakness (3)

6. Refer to weakness (5)

**Reasons To Accept:**

1. This paper is the first to combine TKGs with N-tuple KGs resulting in NTKGs and it also provides two NTKG datasets for evaluations.

2. The proposed NE-Net considers and makes use of the main characteristics of TKG extrapolation and N-tuple KG reasoning.

**Reasons To Reject:**

1. The paper is not easy to follow especially in section 4. In section 4.1.1, the author mentioned “predicate instance” and “predicate type” which are not properly defined or illustrated and as a result, makes the method introduction really hard to follow.

2. I think the results are not convincible enough. Some recent and competitive TKG extrapolation models and N-tuple KG reasoning models like: TLogic, xERTE, CENET, Hyper2 and GRAN are not included.

3. I think the necessity of introducing NTKG reasoning models is not well supported. Because, for N-tuple KGs, they can represent temporal facts by adding a role “timestamp” and the corresponding value is the date. In this way, static N-tuple models can handle temporal N-ary facts. Thus, when evaluating static N-tuple models, except for the trivial experiments the author having done in this paper, treating timestamps of facts in NTKGs as an auxiliary role-value pairs and then running static models on them would make more sense.

4. One of the motivations of proposing NE-Net is to model the temporal patterns of historical facts. However, in section 5.1.4, the window m is to only 2 which does not comply with the motivation and makes the utilization of temporal information questionable.

5.It is unclear how many parameters this complex model uses and how it compares with existing approches.

**Reproducibility:**

3: Could reproduce the results with some difficulty. The settings of parameters are underspecified or subjectively determined; the training/evaluation data are not widely available.

**Reviewer Confidence:**

4: Quite sure. I tried to check the important points carefully. It's unlikely, though conceivable, that I missed something that should affect my ratings.

---

> ### Author Rebuttal · Authors · 2023-08-29
>
> Thank you for the effort you have invested in reviewing our work. We are honored to address the concerns you've raised.
>
> **Q1**: The paper is not easy to follow, especially in section 4. In section 4.1.1, the author mentioned “predicate instance” and “predicate type” which are not properly defined or illustrated and as a result, makes the method introduction really hard to follow. What is a predicate instance and predicate type?
>
> **A1**: The predicate type is a concept in the knowledge base, which becomes a predicate instance when its arguments are specified [1][2]. For example, “(attack, attacker: person A, victim: person B)” and “(attack, attacker: person C, victim: person D)” are two distinct predicate instances. However, they both belong to the "attack" predicate type. We will describe these two terms more clearly in the latter version.
>
> **Q2**: The results are not convincible enough. Some recent and competitive TKG extrapolation models and N-tuple KG reasoning models like: TLogic, xERTE, CENET, Hyper2 and GRAN are not included. Especially, the improvements on NICE are marginal.
>
> **A2**: Refer to **the performance of the competitive TKG extrapolation models**, we compare our model with three new and representative TKG reasoning methods, namely GHT [3], CEN [4], and TiRGN [5]. The corresponding results are as follows:
>
> |       | NWIKI | NWIKI | NWIKI | NWIKI | NWIKI | NWIKI | NWIKI | NWIKI |  NICE | NICE |  NICE | NICE |  NICE | NICE |  NICE | NICE |
> |:-----:|:-----:|:-----:|:-----:|:-----:|:-----:|:-----:|:-----:|:-----:|:-----:|:----:|:-----:|:----:|:-----:|:----:|:-----:|:----:|
> |       |  H@1  |  H@1  |  H@3  |  H@3  |  H@10 |  H@10 |  MRR  |  MRR  |  H@1  |  H@1 |  H@3  |  H@3 |  H@10 | H@10 |  MRR  |  MRR |
> | Model |   C   |  AUX  |   C   |  AUX  |   C   |  AUX  |   C   |  AUX  |   C   |  AUX |   C   |  AUX |   C   |  AUX |   C   |  AUX |
> |  GHT  | 30.71 |   -   | 37.78 |   -    | 39.94 |    -   | 34.57 |   -    | 30.28 |   -   | 45.20 |   -   | 61.04 |   -   | 40.61 |   -   |
> |  CEN  | 48.47 |   -    | 62.97 |   -    | 70.10 |   -    | 56.89 |    -   | 33.32 |   -   | 49.29 |   -   | 64.65 |    -  | 43.98 |   -   |
> | TiGRN | 50.61 |   -    | 68.24 |    -   | 81.13 |   -    | 61.10 |    -   | 34.82 |   -   | 51.54 |   -   | 66.47 |    -  | 45.77 |   -   |
>
> As we can see, NE-Net still outperforms these baselines across all metrics on both datasets. We will manage to add other missing baselines and references in a later version.
>
> Refer to **the marginal improvements on NICE**, this may be caused by the limited auxiliary role types in NICE. The detailed analysis is provided in lines 578-583.
>
> **Q3**: The necessity of introducing NTKG reasoning models is not well supported. Because, for N-tuple KGs, they can represent temporal facts by adding a role “timestamp” and the corresponding value is the date. In this way, static N-tuple models can handle temporal N-ary facts. Thus, when evaluating static N-tuple models, except for the trivial experiments the author having done in this paper, treating timestamps of facts in NTKGs as auxiliary role-value pairs and then running static models on them would make more sense.
>
> **A3**: To verify the capability of existing N-tuple KG models in performing temporal reasoning tasks, we have conducted supplementary experiments following the suggested setting. Specifically, we kept the time information in n-tuple facts, and retrained NALP, HINGE, as well as Hy-Transformer on NWIKI for predicting the core entities. The results are presented in the following table, where the models suffixed with "w. time" indicating the training process with time information:
>
> | model\metrics | H@1 | H@3 | H@10 | MRR |
> |:---:|:---:|:---:|:---:|:---:|
> | NALP | 10.59 | 10.59 | 22.52 | 14.86 |
> | NALP w. time | 11.91 | 14.62 | 18.36 | 14.16 |
> | HINGE | 19.10 | 23.47 | 25.91 | 21.74 |
> | HINGE w. time | 11.59 | 15.63 | 19.71 | 14.49 |
> | Hy-transformer | 33.40 | 35.85 | 37.84 | 34.97 |
> | Hy-transformer w. time | 29.93 | 32.25 | 34.05 | 31.42 |
>
> We observed that these existing N-tuple KG reasoning methods have shown comparable, and even worse performance when compared to the performance not considering time information. Since the N-tuple KG reasoning approaches view the timestamp information as regular entities, the significant temporal order characteristic is omitted. On the contrary, the proposed NE-Net takes the temporal order into consideration, and experimental results verify its effectiveness in the temporal reasoning task.
>
> **Q4**: One of the motivations of proposing NE-Net is to model the temporal patterns of historical facts. The window m is to only 2 which does not comply with the motivation and makes the utilization of temporal information questionable.
>
> **A4**: We performed grid search on the length of time window (1, 6). The optimal lengths of history $m$ on both datasets are experimentally set to 2. The value of $m$ is not too large, as the facts at the most recent timestamps carry greater weight. This is substantiated by several quadruple-based TKG reasoning models [4][6] which confirm that the optimal historical length should not be excessively large in TKG reasoning datasets. For instance, RE-GCN demonstrates an optimal history lengths of 1 and 3 in the ICEWS14 and WIKI datasets, which are constructed from the same source data as the proposed NWIKI and NICE datasets [6]. We provide an ablation study to prove the effectiveness of using temporal information. Specifically, we conduct experiments only using the score function with the randomly initialized learnable embeddings without using history. The corresponding MRR results are as follows:
>
> |  | NWIKI | NWIKI | NWIKI | NICE | NICE | NICE |
> |:---:|:---:|:---:|:---:|:---:|:---:|:---:|
> | variants | C | AUX | ALL | C | AUX | ALL |
> | NE-Net (no his) | 27.30 | 12.80 | 19.01 | 43.90 | 78.40 | 62.69 |
> | NE-Net (w. his) | 72.03 | 57.68 | 63.82 | 48.98 | 79.06 | 65.37 |
>
> The performance gap verifies the effectiveness of the proposed NE-Net using temporal information.
>
> **Q5**: It is unclear how many parameters this complex model uses and how it compares with existing approaches.
>
> **A5**: With regards to the number of parameters, the proposed NE-Net consists of 15.8M weights on the NWIKI, while REGCN, one of the most related baselines, consists of 10.2M weights. Such different scales of parameters can be attributed to variations in the GCN layer configurations and the choice of decoders in each model.
>
> **Q6**: In line 500, the authors mentioned they use the time filter protocol for evaluation. I’d like to know more about the time-aware filtering protocol implemented in NTKGs. For example, in core entity prediction, does it filter out facts with the same core entity, predicate and timestamps or it filter out facts whose entities, predicate, and timestamp are completely the same as facts in datasets? How it is implemented during AUX prediction?
>
> **A6**: In the core entity prediction, the facts having completely the same elements as the query fact except for the query elements are filtered out. Take a typical query $(pred, role_1: e_1, role_2: ?, role_3:e_3, t)$ with answer $o_1$ in the test set as an example, and assume there are two other facts occurring at timestamp $t$, namely, $f_1 = (pred, role_1: e_1, role_2: o_2, role_3:e_3, t)$ and $f_2 = (pred, role_1: e_1, role_2: o_3, role_3:e_4, t)$, only $o_2$ in $f_1$ can perfectly match the query. Therefore, $f_1$ is the only one that is filtered out. The same filtering process applies to the auxiliary entity prediction.
>
>
> [1] Finch S E, Finch J D, Huryn D, et al. An approach to inference-driven dialogue management within a social chatbot[J]. arXiv preprint arXiv:2111.00570, 2021.
>
> [2] Leppanen M. Towards an abstraction ontology[J]. Frontiers in Artificial Intelligence and Applications, 2007, 154: 166.
>
> [3] Sun H, Geng S, Zhong J, et al. Graph Hawkes Transformer for Extrapolated Reasoning on Temporal Knowledge Graphs[C]//Proceedings of the 2022 Conference on Empirical Methods in Natural Language Processing. 2022: 7481-7493.
>
> [4] Li Z, Guan S, Jin X, et al. Complex Evolutional Pattern Learning for Temporal Knowledge Graph Reasoning[C]//Proceedings of the 60th Annual Meeting of the Association for Computational Linguistics (Volume 2: Short Papers). 2022: 290-296.
>
> [5] Li Y, Sun S, Zhao J. Tirgn: time‐guided recurrent graph network with local‐global historical patterns for temporal knowledge graph reasoning[C]//Proceedings of the Thirty‐First International Joint Conference on Artificial Intelligence, IJCAI 2022, Vienna, Austria, 23‐29 July 2022. ijcai. org, 2022: 2152-2158.
>
> [6] Li Z, Jin X, Li W, et al. Temporal knowledge graph reasoning based on evolutional representation learning[C]//Proceedings of the 44th international ACM SIGIR conference on research and development in information retrieval. 2021: 408-417.

---

### Official Review · Reviewer_NCMC · 2023-08-03

**Soundness:** 3

**Excitement:**

3: Ambivalent: It has merits (e.g., it reports state-of-the-art results, the idea is nice), but there are key weaknesses (e.g., it describes incremental work), and it can significantly benefit from another round of revision. However, I won't object to accepting it if my co-reviewers champion it.

**Missing References:**

References after 2021. For example:

Static n-ary knowledge graph reasoning methods:

PolygonE: Modeling N-ary Relational Data as Gyro-Polygons in Hyperbolic Space


TKG reasoning methods:

https://aclanthology.org/2022.findings-emnlp.458.pdf

https://aclanthology.org/2022.findings-emnlp.542.pdf

**Paper Topic And Main Contributions:**

The paper proposes N-tuple augmented temporal knowledge graph (N-TKGs) to accurately describe the temporal fact of TKGs. The paper proposes a NE-Net to model the interaction among concurrent facts and temporal patterns of N-TKGs. The experiments on predicate prediction and entity prediction demonstrate the effectiveness of the method and outperform the static n-tuple baselines and temporal quadruple baselines.

**Questions For The Authors:**

1. See Reasons to Reject 3. How is the role-entity pair constructed?

2. How do the TKG reasoning baselines reason with full auxiliary information (line 531)? Is the auxiliary entity modeled the same as the core entity? If so, is it possible for them to predict auxiliary entities?

**Reasons To Accept:**

1. The n-tuple augmented TKGs are novel and complement the existing research.

2. The paper is well-written.

3. The experiments on two datasets demonstrate the effectiveness of NE-Net.

**Reasons To Reject:**

1. There are no baselines and references in related works after 2021. The baselines may be outdated.

2. The experiments lack case studies. There is only the analysis of final performance with varying percentages of auxiliary entities. A case study would explain how auxiliary entities help with the reasoning.

3. The dataset construction is not clearly explained. For example, in 5.1.1, how is the (3) step for NWIKI conducted? How is the "type-specific role-entity pair and the occurrence place" obtained in the (2) step for NICE?

**Reproducibility:**

4: Could mostly reproduce the results, but there may be some variation because of sample variance or minor variations in their interpretation of the protocol or method.

**Reviewer Confidence:**

4: Quite sure. I tried to check the important points carefully. It's unlikely, though conceivable, that I missed something that should affect my ratings.

---

> ### Author Rebuttal · Authors · 2023-08-29
>
> Thank you for the effort you have invested in reviewing our work. We are honored to address the concerns you've raised.
>
> **Q1**: There are no baselines and references in related works after 2021. The baselines may be outdated.
>
> **A1**: Thank you for this valuable suggestion. We compare our model with three new and representative TKG reasoning methods, namely GHT [1], CEN [2], and TiRGN [3]. The corresponding results are as follows:
>
> |       | NWIKI | NWIKI | NWIKI | NWIKI | NWIKI | NWIKI | NWIKI | NWIKI |  NICE | NICE |  NICE | NICE |  NICE | NICE |  NICE | NICE |
> |:-----:|:-----:|:-----:|:-----:|:-----:|:-----:|:-----:|:-----:|:-----:|:-----:|:----:|:-----:|:----:|:-----:|:----:|:-----:|:----:|
> |       |  H@1  |  H@1  |  H@3  |  H@3  |  H@10 |  H@10 |  MRR  |  MRR  |  H@1  |  H@1 |  H@3  |  H@3 |  H@10 | H@10 |  MRR  |  MRR |
> | Model |   C   |  AUX  |   C   |  AUX  |   C   |  AUX  |   C   |  AUX  |   C   |  AUX |   C   |  AUX |   C   |  AUX |   C   |  AUX |
> |  GHT  | 30.71 |   -   | 37.78 |   -    | 39.94 |    -   | 34.57 |    -    | 30.28 |   -   | 45.20 |   -   | 61.04 |   -   | 40.61 |   -   |
> |  CEN  | 48.47 |   -    | 62.97 |   -    | 70.10 |   -    | 56.89 |    -   | 33.32 |   -   | 49.29 |   -   | 64.65 |    -  | 43.98 |   -   |
> | TiGRN | 50.61 |   -    | 68.24 |    -   | 81.13 |   -    | 61.10 |    -   | 34.82 |   -   | 51.54 |   -   | 66.47 |    -  | 45.77 |   -   |
>
> As we can see, NE-Net still outperforms these baselines across all metrics on both datasets. We will manage to add other missing baselines and references in a later version.
>
> **Q2**: The experiments lack case studies. There is only the analysis of final performance with varying percentages of auxiliary entities. A case study would explain how auxiliary entities help with the reasoning.
>
> **A2**: Thank you for your constructive comment. We show two cases in the following table, where NE-Net correctly predicts the answer entity. In this table, $t-1$ and $t$ represent historical timestamps, while $t+1$ represents the query timestamp. In the first case, the correct answer to the query plays as an auxiliary entity in the latest history fact. The model is hard to predict the correct answer without such auxiliary information. In the second case, the auxiliary entity "Humanitarian Aid" helps the model establish a connection between the query and the most informative historical fact, and thereby enabling more accurate predictions.
>
> | Case number | History at t-1 | History at t | Query at t+1 | Answer |
> |:---:|:---|:---|:---|:---:|
> | Case 1 | (Disapprove, Opponent: Barack Obama,   Proponent: Head of Government (Kenya),   Disapprove Way: Denounce,    Place: Kenya) | (Consult,      Consulter: Barack Obama,     Consulted: Mulatu Teshome,     Consult Way: Engage in Negotiation,     Place: Ethiopia)  | (Engage in Diplomatic Cooperation,     Cooperator: Barack Obama,     Cooperator: ?,     Cooperation way: Praise or Endorse) | Ethiopia |
> | Case 2 | (Express Intent to Cooperate,     Volunteer: Iran,     Cooperation Target: South Africa, Cooperate Content: Engage in Diplomatic Cooperation); (Express Intent to Cooperate,     Volunteer: Iran,      Cooperation Target: Yemen, Cooperate Content: Provide Humanitarian Aid) | (Engage in Diplomatic Cooperation,     Cooperator: Iran,    Cooperator: South Africa,     Cooperation way: Sign Formal Aggrement) | (Provide Aid,     Provider: Iran,     Recipient: ?,     Aid Content: Humanitarian Aid) | Yemen |
>
> **Q3**: The dataset construction is not clearly explained. For example, in 5.1.1, how is the (3) step for NWIKI conducted? How is the "type-specific role-entity pair and the occurrence place" obtained in the (2) step for NICE? How is the role-entity pair constructed?
>
> **A3**: Refer to the question **“how is the third step in NWIKI conducted”**: During the building process of NWIKI, we filtered out entities of low frequency, as their behaviors are difficult to learn and predict for most data-driven based methods. Correspondingly, facts involving these entities were also filtered out. After that, we notice that the instances of the "Position Held" predicate comprise over 50% of the dataset. However, the filtering operation in the previous step could impair connectivity. To maintain a robust connection with other predicates, we retained facts that are associated with other predicates and shared core entities with the "Position Held" instances.
>
> Refer to the question **“how to obtain the type-specific role-entity pair and the occurrence place and how to construct the role-entity pair in NICE”**: There is partial auxiliary information, i.e., the occurrence places, in the raw event records provided by ICEWS. And we observed that the predicate names of ICEWS contain additional auxiliary information. For example, the predicate names “Cooperate Economically” and “Engage in Judicial Cooperation” contain the specified cooperation aspects of the general predicate “Engaged in Cooperation”, namely “Economic” and “Militarily”, respectively. To obtain such auxiliary information, we designed rule templates to extract them from the specified predicate names and merged these specified predicates into the general ones. Since role information is absent in the raw event records, we manually assigned role names to each predicate.
>
> **Q4**: How do the TKG reasoning baselines reason with full auxiliary information (line 531)? Is the auxiliary entity modeled the same as the core entity? If so, is it possible for them to predict auxiliary entities?
>
> **A4**: We apologize for omitting details about the baseline reproduction process. As illustrated in the paper, we converted the concurrent facts in the history into graph sequences, where the predicate instances are treated as nodes. Such operation is similar to the CVT node in knowledge graphs [4] and allows each predicate instance to connect with various auxiliary entities. By decomposing the predicate instances of each graph into multiple quadruples, each auxiliary entity therein can be modeled in the same way as that of the core entity, and thus existing TKG reasoning methods can predict auxiliary entities. However, we observe that including the auxiliary entities for these methods will lead to worse performance on core entity prediction. For example, the MRR result of REGCN on the core entity prediction decreases from 56.78 to 21.52. We guess that it is because the future predicate instances are unseen and cannot be learned since each predicate instance only appears once. As a result, we abandon the prediction of auxiliary entities and focus on the prediction of core entities. We will report this part of results in the next version.
>
> [1] Sun H, Geng S, Zhong J, et al. Graph Hawkes Transformer for Extrapolated Reasoning on Temporal Knowledge Graphs[C]//Proceedings of the 2022 Conference on Empirical Methods in Natural Language Processing. 2022: 7481-7493.
>
> [2] Li Z, Guan S, Jin X, et al. Complex Evolutional Pattern Learning for Temporal Knowledge Graph Reasoning[C]//Proceedings of the 60th Annual Meeting of the Association for Computational Linguistics (Volume 2: Short Papers). 2022: 290-296.
>
> [3] Li Y, Sun S, Zhao J. Tirgn: time‐guided recurrent graph network with local‐global historical patterns for temporal knowledge graph reasoning[C]//Proceedings of the Thirty‐First International Joint Conference on Artificial Intelligence, IJCAI 2022, Vienna, Austria, 23‐29 July 2022. ijcai. org, 2022: 2152-2158.
>
> [4] Wen J, Li J, Mao Y, et al. On the representation and embedding of knowledge bases beyond binary relations[J]. arXiv preprint arXiv:1604.08642, 2016.

---

### Official Review · Reviewer_vaso · 2023-08-05

**Typos Grammar Style And Presentation Improvements:** 1. Table 3
**Soundness:** 4

**Excitement:**

3: Ambivalent: It has merits (e.g., it reports state-of-the-art results, the idea is nice), but there are key weaknesses (e.g., it describes incremental work), and it can significantly benefit from another round of revision. However, I won't object to accepting it if my co-reviewers champion it.

**Missing References:**

1. Xu, Yi, et al. "Temporal knowledge graph reasoning with historical contrastive learning." Proceedings of the AAAI Conference on Artificial Intelligence. Vol. 37. No. 4. 2023.

2. Li, Yujia, Shiliang Sun, and Jing Zhao. "Tirgn: time‐guided recurrent graph network with local‐global historical patterns for temporal knowledge graph reasoning." Proceedings of the Thirty‐First International Joint Conference on Artificial Intelligence, IJCAI 2022, Vienna, Austria, 23‐29 July 2022. ijcai. org, 2022.

3. ...

**Paper Topic And Main Contributions:**

The paper proposes a new temporal KG data structure: N-tuple, two curated N-tuple-based datasets, and a sophisticated reasoning model NE-Net. The experimental results show its superiority over existing models.

**Questions For The Authors:**

1. In Section 5.1.1, why did you filter out the entities of low frequency?

2. Since NE-Net models include recurrent modeling (Modeling Interactions in Temporal Adjacent Facts) and Transformer modules, what is the time complexity of NE-Net?

**Reasons To Accept:**

1. The paper holds the view that the existing quadruple-based formulation cannot describe temporal facts accurately, thus limiting the applications of TKGs, which is a new perspective of TKG reasoning.

2. The authors build two new N-tuple-based datasets to support its claims and model, which will be of some help to the TKG community.

**Reasons To Reject:**

1. The claim that existing quadruple-based formulation cannot describe temporal facts accurately is not convincing, which means the papers should provide experiments to compare how the N-tuple-based structure is better than the quadruple-based structure to support the claim.

2. The newly constructed datasets determine the upper border of model performance, but the descriptions of how to construct these two datasets are not clear, especially for the selection preference of predicates as the number of predicates is too small (22 for NWIKI, 20 for NICE).

3. Some SOTA baselines published in early 2023 are missing (see Missing References section).

**Reproducibility:**

4: Could mostly reproduce the results, but there may be some variation because of sample variance or minor variations in their interpretation of the protocol or method.

**Reviewer Confidence:**

5: Positive that my evaluation is correct. I read the paper very carefully and I am very familiar with related work.

---

> ### Author Rebuttal · Authors · 2023-08-29
>
> Thank you for the effort you have invested in reviewing our work. We are honored to address the concerns you've raised.
>
> **Q1**: The claim that existing quadruple-based formulation cannot describe temporal facts accurately is not convincing, which means the papers should provide experiments to compare how the N-tuple-based structure is better than the quadruple-based structure to support the claim.
>
> **A1**: Thank you for this valuable suggestion. As shown in the last block of Table 3, we present the results of NE-Net with different ratios of auxiliary information (0%, 30%, 60%, 100%). NE-Net (0%) denotes the NE-Net that models quadruple-based facts, while NE-Net (100%) denotes the NE-Net that models N-tuple-based facts (taking into account all auxiliary information considered). The superiority of NE-Net (100%) comparing to NE-Net (0%) justifies the necessity of introducing N-tuple based temporal facts. Furthermore, it can be noted that as the ratio of auxiliary information increases, so does the performance, further highlighting the superiority of the N-tuple-based structure.
>
> To clarify the aforementioned discussion, we will reorganize Table 3 and Section 5.2.2. Specifically, we will segregate the results in the last block into a separate table and provide a more comprehensive analysis of the results in the paper.
>
> **Q2**: (1) The newly constructed datasets determine the upper border of model performance, but the descriptions of how to construct these two datasets are not clear, especially for the selection preference of predicates as the number of predicates is too small (22 for NWIKI, 20 for NICE). (2) In Section 5.1.1, why did you filter out the entities of low frequency?
>
> **A2**: We reorganized the construction process of the dataset and rewrote it as follows:
>
> The NWIKI dataset was constructed as follows: (1) We downloaded the Wikidata dump until November 20, 2017, and extracted the facts which contain the timestamp information and involve human entities; (2) To extract n-tuple predicate instances, we selected predicates with more than 3 role types and retained their corresponding predicate instances; (3) We filtered out entities of low frequency, as their behaviors are difficult to learn and predict for most data-driven based methods. Correspondingly, facts involving these entities were also filtered out; (4) The instances of the "Position Held" predicate comprise over 50% of the dataset. However, the filtering operation in the previous step could impair connectivity. To maintain a robust connection with other predicates, we retained facts that are associated with other predicates and share core entities with the "Position Held" instances; (5) Following RE-NET [1], all facts were split into the training set, the validation set, and the test set by a proportion of 80%:10%:10% according to the time ascending order.
>
> The NICE dataset was constructed as follows: (1) We collected the raw event records from ICEWS [2] from Jan 1, 2005 to Dec 31, 2014; (2) From these raw event records, we extracted core entities, timestamps, and partial auxiliary information, i.e., the occurrence places. (3) We observed that the predicate names contain additional auxiliary information. For example, the predicate names “Cooperate Economically” and “Engage in Judicial Cooperation” contain the specified cooperation aspects of the general predicate “Engaged in Cooperation”, namely “Economic” and “Militarily”, respectively. To obtain such auxiliary information, we designed rule templates to extract them from the specified predicate names and merged these specified predicates into the general ones. (4) Considering the absence of role information in the raw event records, we manually assigned role names to each predicate. (5) Similar to NWIKI, NICE was also split into the training set, the validation set, and the test set by a proportion of 80%:10%:10%.
>
> Refer to the question **“the selection preference of predicates as the number of predicates is too small (22 for NWIKI, 20 for NICE)”**: Despite the ample variety of predicates in Wikidata, the number of predicates is limited when it comes to predicates that contain n-tuple (N>3) roles and are temporally associated. The widely used quadruple-based WIKI dataset, derived from the same source, also encompasses only 24 distinct types of predicates [1]. As for the NICE dataset, the third step of the construction process reduces the number of predicates to 20: The predicates in ICEWS systems are organized in a hierarchical manner, with a maximum number of 256 fine-grained types [3]. As we extracted additional auxiliary information from the specified predicate names and merged the corresponding specified predicates into more general ones, there are only 20 kinds of predicates in NICE. For example, we merged the predicates “Cooperate Economically” and “Engage in Judicial Cooperation” into the predicate “Engaged in Cooperation” after the extraction of the role value "Cooperate Aspect”.
>
> Refer to the question **“In Section 5.1.1, why did you filter out the entities of low frequency?”**:
> The presence of low frequency entities poses challenges for data-driven methods to effectively learn and predict their behaviors. Furthermore, these predicate instances involving these entities may influence the evaluation of the model abilities. Similar filtering operation is also adopted in the construction process widely used TKG datasets, YAGO and WIKI [4].
>
> **Q3**: Some SOTA baselines published in early 2023 are missing (see Missing References section).
>
> **A3**: Thank you for this valuable suggestion. We compare the proposed NE-Net with three new and representative TKG reasoning methods, namely GHT [5], CEN [6], and TiRGN [7]. The corresponding results are as follows:
>
> |       | NWIKI | NWIKI | NWIKI | NWIKI | NWIKI | NWIKI | NWIKI | NWIKI |  NICE | NICE |  NICE | NICE |  NICE | NICE |  NICE | NICE |
> |:-----:|:-----:|:-----:|:-----:|:-----:|:-----:|:-----:|:-----:|:-----:|:-----:|:----:|:-----:|:----:|:-----:|:----:|:-----:|:----:|
> |       |  H@1  |  H@1  |  H@3  |  H@3  |  H@10 |  H@10 |  MRR  |  MRR  |  H@1  |  H@1 |  H@3  |  H@3 |  H@10 | H@10 |  MRR  |  MRR |
> | Model |   C   |  AUX  |   C   |  AUX  |   C   |  AUX  |   C   |  AUX  |   C   |  AUX |   C   |  AUX |   C   |  AUX |   C   |  AUX |
> |  GHT  | 30.71 |   -   | 37.78 |   -    | 39.94 |    -   | 34.57 |    -    | 30.28 |   -   | 45.20 |   -   | 61.04 |   -   | 40.61 |   -   |
> |  CEN  | 48.47 |   -    | 62.97 |   -    | 70.10 |   -    | 56.89 |    -   | 33.32 |   -   | 49.29 |   -   | 64.65 |    -  | 43.98 |   -   |
> | TiGRN | 50.61 |   -    | 68.24 |    -   | 81.13 |   -    | 61.10 |    -   | 34.82 |   -   | 51.54 |   -   | 66.47 |    -  | 45.77 |   -   |
>
> As we can see, NE-Net still outperforms these baselines across all metrics on both datasets. We will manage to add other missing baselines and references in a later version.
>
> **Q4**: Since NE-Net models include recurrent modeling (Modeling Interactions in Temporal Adjacent Facts) and Transformer modules, what is the time complexity of NE-Net?
>
> **A4**: To illustrate the efficiency of the proposed NE-Net, we analyze the computational complexity of its evolutional interaction encoder. We view the computational complexities of the interaction aggregation unit as constants. The time complexity of the EP unit is $O(\omega_1|\mathcal{E}_1|)$, where $|\mathcal{E}_1|$ is the maximum number of role edges in the concurrent facts. The time complexity of the CE unit is $O(\omega_2|\mathcal{E}_2|)$, where $\mathcal{E}_2$ denotes the maximum number of concurrent facts in the history. As we recurrently model on historical KGs at adjacent timestamps for $m$ times, the time complexity for the evolutional interaction encoder is finally $O(m\omega_1|\mathcal{E}_1|+m\omega_2|\mathcal{E}_2|)$.
>
> [1] Jin W, Qu M, Jin X, et al. Recurrent event network: Autoregressive structure inference over temporal knowledge graphs[J]. arXiv preprint arXiv:1904.05530, 2019.
>
> [2] Ward M D, Beger A, Cutler J, et al. Comparing GDELT and ICEWS event data[J]. Analysis, 2013, 21(1): 267-297.
>
> [3] Halterman A, Bagozzi B, Beger A, et al. PLOVER and POLECAT: A New Political Event Ontology and Dataset[J]. 2023.
>
> [4] Dasgupta S S, Ray S N, Talukdar P. Hyte: Hyperplane-based temporally aware knowledge graph embedding[C]//Proceedings of the 2018 conference on empirical methods in natural language processing. 2018: 2001-2011.
>
> [5] Sun H, Geng S, Zhong J, et al. Graph Hawkes Transformer for Extrapolated Reasoning on Temporal Knowledge Graphs[C]//Proceedings of the 2022 Conference on Empirical Methods in Natural Language Processing. 2022: 7481-7493.
>
> [6] Li Z, Guan S, Jin X, et al. Complex Evolutional Pattern Learning for Temporal Knowledge Graph Reasoning[C]//Proceedings of the 60th Annual Meeting of the Association for Computational Linguistics (Volume 2: Short Papers). 2022: 290-296.
>
> [7] Li Y, Sun S, Zhao J. Tirgn: time‐guided recurrent graph network with local‐global historical patterns for temporal knowledge graph reasoning[C]//Proceedings of the Thirty‐First International Joint Conference on Artificial Intelligence, IJCAI 2022, Vienna, Austria, 23‐29 July 2022. ijcai. org, 2022: 2152-2158.

---

### Meta-Review · Area_Chair_a7Ur · 2023-09-15

**Recommendation:** 3

**Metareview:**

This paper introduces a novel strategy for modelling temporal knowledge graphs, based on n-tuples associated with timestamps. Further, the authors introduce two novel datasets for N-tuple modelling. Experimental results on these new datasets show that the proposed strategy is superior to current alternatives.

While the initial version lacked clarity around certain areas, including the construction process for the two datasets, the authors have addressed this in their rebuttals. The authors have also provided additional evaluation results with more recent baselines, showing continued competitiveness.

One concern mentioned by several reviewers is the necessity of moving to an N-tuple format rather than the previously used quadruple format; although the introduced datasets are useful, and the introduced models outperform contemporaries on these, this argument may not be sufficiently well-supported.

---

### Decision · Program_Chairs · 2023-10-07

**Decision:**

Accept-Findings

**Comment:**

This paper introduces a novel strategy for modelling temporal knowledge graphs, based on n-tuples associated with timestamps. Further, the authors introduce two novel datasets for N-tuple modelling. Experimental results on these new datasets show that the proposed strategy is superior to current alternatives.

While the initial version lacked clarity around certain areas, including the construction process for the two datasets, the authors have addressed this in their rebuttals. The authors have also provided additional evaluation results with more recent baselines, showing continued competitiveness.

One concern mentioned by several reviewers is the necessity of moving to an N-tuple format rather than the previously used quadruple format; although the introduced datasets are useful, and the introduced models outperform contemporaries on these, this argument may not be sufficiently well-supported.